# Financial Market Development and Pollution Nexus in Saudi Arabia: Asymmetrical Analysis

**Haider Mahmood** [1,*] , **Abdullatif Sulaiman Alrasheed** [1] **and Maham Furqan** [2]

1   Department of Finance, College of Business Administration, Prince Sattam bin Abdulaziz University, P.O. Box 165, Al-Kharj 11942, Saudi Arabia; ksu5@hotmail.com
2   S&P Global Marketing Intelligence, Islamabad 44000, Pakistan; mahamfurqan@ymail.com
*   Correspondence: haidermahmood@hotmail.com; Tel.: +966-115-887-037

**Abstract:** The study is aimed to scrutinize the presence of Environmental Kuznets Curve (EKC) hypothesis in Saudi Arabia by analyzing a period of 1971–2014. Asymmetrical impacts of Financial Market Development (FMD) and energy consumption per capita have also been tested on $CO_2$ emissions per capita. The estimates buoyed the long and short-run relationships in the hypothesized model, and EKC is found to be true in terms of the relationship between income and pollution. Asymmetrical effects of FMD in the long run and asymmetrical effects of energy consumption per capita in the long and short run are presented on the $CO_2$ emissions per capita. A decreasing FMD is found responsible for environmental degradation, and decreasing energy consumption per capita is found helpful in controlling $CO_2$ emissions. The tested effect of the financial crisis is found insignificant on $CO_2$ emissions.

**Keywords:** $CO_2$ emissions; Financial Market Development (FMD); Environmental Kuznets Curve (EKC); asymmetrical effects

## 1. Introduction

Economic development tends to take an extended amount of time to occur while its returns and consequences leave long-lasting effects on the overall structure of a country as well. While many economic, financial, social and political aspects get affected by economic development, one effect that cannot be ignored is the alterations it initiates in the environmental profile of the country. For instance, as the Environmental Kuznets Curve (EKC) by Grossman and Krueger [1] argues that when a country moves towards making progress, environmental degradation in the country first takes on a positive trend but decreases later on. Therefore, economic growth may increase pollutant emissions through scale effect in the first phase of EKC. Scale effect explains that with economic growth or rising economic activities, energy consumption may increase which, in turn, increases pollutant emissions. Furthermore, economic growth can lead to a reduction in pollutant emissions if economic growth has matured enough to support clean energy through strict regulations and research and development. This phenomenon is termed as the technique effects. Further, economic growth may result in composition effect through conversion of the dirty sector to clean like the services sector.

Andreoni and Levinson [2] argued that economic growth tends to have an inverted U-shaped relation with deterioration in the environment. Using a static model, it is suggested that pollution abatement tools are used in countries so that the curve can be brought to a downward trend after hitting a peak. Dinda [3] provided a literature review on the concept of EKC and how the idea surfaced and progressed throughout the years. The review offers a consolidated version of a large set of research studies investigated the concept of the association between emissions of a country and economic growth. Both theoretically and practically, the proposition becomes relevant because when a state is

making a way towards financial and economic uplifting, the environment has to take a blow because of the rapid development for some time. However, as the economy keeps pacing rapidly towards sustainable growth, there comes a time when the growth reaches a point promising enough to ensure environmental sustainability.

Along with economic development and increasing energy consumption, many other factors in an economy are said to have a robust influence on environmental degradation; one of these factors discussed in the literature is the influence of Financial Market Development (FMD). Frankel and Romer [4] argued that financial markets might help to achieve a better level of development in a country and demand for a cleaner environment may increase with more development. On the other hand, Zhang [5] argued that FMD expands investment and consumer credit. The increasing economic activities due to FMD may release pollution and become a source of environmental degradation. For economic growth and FMD, overall positive environmental effect may be expected if technique and composition effects are dominant on the scale effect and adverse environmental effect may be expected otherwise.

There is vast environment literature available in the global context but, we limit our discussions to the Middle East and North African (MENA) region because of Saudi Arabia's location. In the MENA and Gulf Cooperation Council (GCC) regions' environment literature, studies have tested the causal relationships in economic growth, energy consumption and $CO_2$ emissions [6,7]. Some other studies have focused on the monotonic and symmetrical effects of Foreign Direct Investment (FDI), energy consumption, economic growth, FMD on the $CO_2$ emissions in different panels of MENA countries [8–10]. Considering the nonlinear effect of economic growth on $CO_2$ emissions in the EKC hypothesis testing, Arouri et al. [11] and Ozcan [12] validated the EKC hypothesis in the MENA countries. On the other hand, Rafindadi et al. [13] could not validate the existence of the EKC hypothesis in the GCC countries. Hence, the validity of EKC is mixed, and investigation of environmental effects of FMD is scant.

In the case of Saudi Arabia, monotonic effects of economic growth and energy consumption are found statistically significant by Alkhathlan and Javid [14] but are found insignificant by Bekhet et al. [15]. Further, Bekhet et al. [15] reported insignificant effects of FMD on the $CO_2$ emissions. On the other hand, Mahalik et al. [16] found a significant effect of FMD on Saudi energy consumption leading to higher emissions since most of the Saudi energy consumption is from fossil fuel. But, they do not test the effects of FMD on pollution emissions. In studies by Alkhathlan and Javid [14] and Bekhet et al. [15], the potential quadratic relationship of economic growth and $CO_2$ emissions is ignored in testing the EKC hypothesis. Although, Alshehry and Belloumi [17] have investigated the EKC hypothesis in the Saudi transport sector. Ignoring the rest of the economy, they could not validate the EKC hypothesis hence this conclusion cannot be generalized for the whole Saudi economy. Therefore, testing of the EKC hypothesis for Saudi Arabia is missing altogether on the aggregated level. Further, insignificant symmetrical effects of FMD and energy consumption on the $CO_2$ emissions reported by Bekhet et al. [15] give direction to test these effects in the asymmetry settings as testing symmetrical effects in the presence of significant asymmetries may be claimed for misspecification of the model [18]. Therefore, our objectives are to investigate the asymmetrical effects of FMD and energy consumption on $CO_2$ emissions and to test the EKC hypothesis at the aggregate level in Saudi Arabia. Testing both objectives may help to fill the Saudi environment literature gap in the current state of the art.

## 2. Literature Review

In the empirical studies, there is a mix of evidence of EKC hypothesis. In their research, Kibria et al. [19] tested the curve and its relevance on a panel dataset of 151 countries from 1971–2013. The conclusion of the study aligned with the Kuznets curve, and they suggested the presence of a polynomial association in the income and fossil fuel share. In the relationship of income and $CO_2$ emissions, Lau et al. [20] and Churchill et al. [21] analyzed a panel of OECD countries and landed

on the same conclusion that these economies are illustrated a mechanism similar to the EKC in play. While, Vlontzos et al. [22] found an N-shape relationship between Gross Domestic Product (GDP) and emissions in EU agriculture sector, Urban and Nordensvard [23] could find the evidence of EKC hypothesis for four out of five Nordic countries. Arango-Miranda et al. [24] could not find the existence of the EKC hypothesis at all in a mixed panel of 10 developed and developing countries. Using a non-parametric model, Azomahou et al. [25] analyzed data of 100 countries from 1960–1996 and proved that the relation is structurally stable between two variables; economic development and $CO_2$ emissions but the EKC hypothesis could not be found true.

In the solution of environmental effects of growth, Wang et al. [26] mentioned that with governments implementing restrictions on carbon emissions and related activities, it has started to become a common practice among nations to outsource these activities that generate high carbon emissions. In this way, carbon-curbing requirements of countries are met. In these practices, retail-undertaking contract yields are said to be the best options for good performance and higher profits. Turki et al. [27] talked about the idea of manufacturing and remanufacturing systems with storage facilities that take place under any carbon reduction and trade-related policies. It is mentioned that remanufacturing is becoming a popular trend in the current light of globalization as companies are making efforts to make their systems more efficient. Three main factors including set-up-cost, returned used item percentage and availability of machinery put an impact on the manufacturing and remanufacturing. Higher carbon trading price and lower caps on carbon emissions also help the producers to look for low carbon production strategies and to seek remanufacturing options.

As countries start to make economic progress, it becomes easier for them to incorporate cleaner ways of production and cut down on emissions. These cleaner ways of production include installation of advanced technologies that generate fewer carbon emissions while keeping the production level at the same point. However, even if adopting newer and cleaner technologies is possible for economically stable countries, they might face challenges due to lags in social development. Boerenfijn et al. [28] discussed the case of the Netherlands where the increasing older population is leading to higher housing as well as energy demands. The social housing associations in the country are making efforts to achieve housing demands while keeping the energy goals into account. McCabe et al. [29] discussed a similar idea where individuals and societies together play a role in energy transition and the development of renewable energy.

A stream of literature follows the STIRPAT model in the estimations. For example, Sun et al. [30] investigated the determinants of $CO_2$ emissions from the power sector of China. They found the two most important factors which may help in pollution reduction that are electricity production with low carbon technology and economic activities. Cui et al. [31] compared the determinants of $CO_2$ emissions from power sector in China. They find that economic growth and population have contributed most in the $CO_2$ emissions and electricity intensity and industrial structure contribute least in the $CO_2$ emissions. Wang et al. [32] investigated the determinants of household $CO_2$ emissions using 30 provinces' data from 2006–2015. It was found that electricity and coal consumptions for food and transport are responsible for household $CO_2$ emissions. Further, a number of households, demographic factors, carbon intensity and GDP per capita were also found responsible for $CO_2$ emissions. On the other hand, the industrial structure was seen to assist in reducing $CO_2$ emissions.

In the FMD and $CO_2$ emissions relationship, Shahbaz et al. [33] reported that FMD had been found helpful in reducing $CO_2$ emissions in Malaysia. However, income and energy consumption have been found responsible for environmental degradation. On the other hand, Shahbaz et al. [34] found that FMD, along with income and energy consumption, have destroyed the environment in Pakistan. Paramati et al. [35] tested the influence of stock market growth and FDI on the emissions and argued that these variables have a substantial long-term effect on $CO_2$ emissions of G20 countries. Growing the interest in the environmental related studies, a number of studies have been focused in the MENA region. For instance, Arouri et al. [11] investigated the presence of EKC for the 12 MENA countries using a period 1981–2005 and find an existence of EKC hypothesis through a quadratic effect of income

on $CO_2$ emissions. Further, they discovered that energy consumption is responsible for increasing $CO_2$ emissions. Using data of 1980–2009, Al-Mulali [8] conducted a study on MENA countries and researched how the level of oil consumption and economic growth put an impact on the $CO_2$ emissions of the region. The results of the panel study aligned with the hypothesis that increasing income results in more energy consumption which adversely affects the $CO_2$ emissions. Omri [6] investigated the 14 MENA countries using a period 1990–2011 and reported a bidirectional connection between income and $CO_2$ emissions and a unidirectional relationship from energy usage to $CO_2$ emissions.

In the panel of Middle East, Al-Mulali [9] has analyzed 12 countries of a period from 1990–2009. He aimed to investigate how FDI, GDP, energy consumption and $CO_2$ emissions are inter-related. In the empirical results, it was found that all the factors seem to have a strong influence on the $CO_2$ emission and pollution portfolio of the Middle East. While FDI financially benefits the countries, there is a need to develop sophisticated energy efficiency mechanisms so that the blow of economic progress does not make the pollution worse in the region. In EKC hypothesis testing, Ozcan [12] worked on 12 Middle East countries and reported that an U-shape relationship exists between income and $CO_2$ emissions in the case of five countries, an inverted U-shape relationship in the case of three countries and no relationship has been found for the rest of the countries. Further, energy consumption is found to be responsible for environmental degradation in the case of most of the investigated countries. In the panel causality results, unidirectional causality is only found from income and its square to energy consumption.

In a more narrowed down Gulf Cooperation Council (GCC) region, Qader [36] claimed that most of the pollution emissions are from the oil drilling and electricity production. When it comes to energy conservation policies, an idea provided by Al-Mulali and Ozturk [7] is that energy policy can have a robust influence on the overall economic and environmental health of countries in the GCC region. They applied the time series techniques on the six GCC countries' models and found that fossil fuel electricity consumption and income variable have a bidirectional relationship in some GCC countries. Therefore, increasing electricity consumption due to higher levels of economic growth may have harmful environmental effects. In the panel of GCC countries using a period of 1980–2012, Salahuddin et al. [10] found that energy usage and income are contributing in the $CO_2$ emissions. It is because when a country makes its way up the economic ladder, it tends to use energy especially electricity way more, which then leads to a higher energy generation, hence emitting higher pollution. There is a need for GCC countries to focus on technologies that promote clean energy generation and do not affect the environment in a degrading way.

Using a period 1990–2014, Rafindadi et al. [13] studied the effect of FDI on pollution in a panel of GCC region countries while testing the EKC hypothesis. The presence of the EKC hypothesis could not be found, and energy consumption was found responsible for higher $CO_2$ emissions. Further, domestic and foreign investments play a positive environmental role by reducing the $CO_2$ emissions. The analysis has led to the derivation of the conclusion that when a country in the GCC region receives a larger inflow of FDI, there is a higher chance to protect from environmental degradation through clean technologies. Salahuddin et al. [37] mentioned that in Kuwait, positive contributors to $CO_2$ emissions are income, energy usage and FDI and these effects sustain both in short and long runs. It is suggested that it is crucial for Kuwait to come up with policies that can help to reduce emissions. Some of these policies might be around carbon capture, moving to solar and wind, reducing subsidy on residential electricity and installation of storage plants to save energy. However, the effect of FMD is found negative, so FMD has positive environmental effects in Kuwait.

In the case of Saudi Arabia, Alkhathlan and Javid [14] investigated the monotonic effects of income and petroleum, gas and electricity consumption on the $CO_2$ emissions using a period 1980–2011. They found that all types of energy usage have adverse environmental effects in both long and short runs. Further, the monotonic effect of income was found positive in most of the investigated models. Alshehry and Belloumi [17] investigated the EKC hypothesis for a relationship of transport $CO_2$ emissions and income in Saudi Arabia using data from 1971–2011. The inverted U-shaped association

was not found which confirm the non-existence of EKC. However, they have confirmed the intertwining relationship of pollution and energy usage in causality analyses. Using a limited period 1980–2011, Bekhet et al. [15] examined the monotonic environmental effects of income, energy usage and FMD for each GCC country. In the case of Saudi Arabia, they reported all insignificant effects in the long run. However, they reported a significant causality from $CO_2$ emissions to the energy usage and FMD, but the inverse evidence was not found.

Using a period 1971–2011, Mahalik et al. [16] explored the quadratic effect of FMD on the energy usage in Saudi Arabia and found an inverted U-shape relationship. Further, they have reported the uni-directional causality from financial development to energy demand which may have consequent environmental effects. In the analysis of the symmetrical effect of FMD on $CO_2$ emissions, Bekhet et al. [15] could not establish the significant relationship due to assuming symmetrical effect, linear effect of income and limited time sample. Further, Alshehry and Belloumi [17] could not establish the EKC hypothesis in Saudi Arabia which may be again an outcome of misspecification of a model. With a lot of energy sector and fossil fuel energy consumption, it is crucial to understand how the energy consumption and the FMD of the region are not associated with pollution emissions in the past literature.

Mahmood and Alkhateeb [18] claimed that the results of the symmetrical investigation, in the presence of significant asymmetry in the model, may become a reason of misspecification of the model and may as a result generate biased estimates. Although the asymmetrical effects of FMD and income on Pakistani $CO_2$ emissions have been tested in [34] by ignoring the testing of the EKC hypothesis, this current study extends the scope of the model by analyzing asymmetrical effects of FMD and energy consumption along with testing the EKC hypothesis which is missing by [34]. Therefore, considering both asymmetrical effects and EKC may be claimed unique and a contribution in the global environment literature. There is a lack of evidence on the asymmetrical effects of FMD and energy consumption on $CO_2$ emission in the Saudi literature and testing of the EKC hypothesis at the aggregate level is also missing. These are the potential research gaps which require significant attention and are targeted by this present research to fill. Due to the current position of Saudi Arabia in the world oil market, its emissions and economic scenario have a potentially global impact. Hence, it is crucial to understand how EKC applies in the country and what the asymmetrical effects of FMD and energy consumption are.

## 3. Methods

### 3.1. Data and Source

Table 1 shows the definition of all utilized variables. Aggregate data of Saudi Arabia on each variable was extracted for each year from the World Development Indicators (WDI) and is available as Supplementary Material. We restrict analysis up to 2014 as pollution and energy variables are not available after 2014. Further, we utilize the same source to keep homogeneity in the all mentioned definitions of variables in Table 1 throughout the utilized period 1971–2014.

**Table 1.** Description of Variables.

| Variable | Description |
| --- | --- |
| $CO_t$ | Natural Logarithm of $CO_2$ emissions metric tons per capita which is proxy for pollution emissions |
| $Y_t$ | Gross Domestic Product (GDP) per capita in constant US dollars |
| $Y_t^2$ | A square term of $Y_t$ |
| $FM_t$ | A percentage of domestic credit to GDP which is a proxy for financial market development |
| $PFM_t$ | Partial sum of positive changes in $FM_t$ |
| $NFM_t$ | Partial sum of negative changes in $FM_t$ |
| $EC_t$ | Energy consumption kg of oil equivalent per capita |
| $PEC_t$ | Partial sum of positive changes in $EC_t$ |
| $NEC_t$ | Partial sum of negative changes in $EC_t$ |
| $t$ | Annual time period from 1971–2014 |

### 3.2. Model and Methodology

In the theoretical prediction discussed in the introduction section, we may suppose the following model:

$$CO_t = f(Y_t, Y_t^2, FM_t, EC_t) \tag{1}$$

$Y_t$ and $Y_t^2$ are included to capture the expected inverted U-shape relation between income and pollution to validate the EKC hypothesis. The estimated positive (negative) coefficients of $Y_t$ ($Y_t^2$) may validate the EKC hypothesis. In Equation (1), we assume a quadratic impact of income per capita on per capita $CO_2$ emissions to validate the EKC in the traditional way. Further, Shahbaz et al. [34] proposed to test the asymmetrical effects of FMD and energy consumption on the $CO_2$ emissions. Following Shin et al. [38], we introduce the nonlinear or asymmetrical effects of $FM_t$ and $EC_t$ by converting each variable into positive and negative variables in the following way:

$$PFM_t = \sum_{i=1}^{t} \Delta FM_i^+ = \sum_{i=1}^{t} \max(\Delta FM_i, 0) \tag{2}$$

$$NFM_t = \sum_{i=1}^{t} \Delta FM_i^- = \sum_{i=1}^{t} \min(\Delta FM_i, 0) \tag{3}$$

$$PEC_t = \sum_{i=1}^{t} \Delta EC_i^+ = \sum_{i=1}^{t} \max(\Delta EC_i, 0) \tag{4}$$

$$NEC_t = \sum_{i=1}^{t} \Delta EC_i^- = \sum_{i=1}^{t} \min(\Delta EC_i, 0) \tag{5}$$

Now, $PFM_t$ ($NFM_t$) are capturing partial sum of positive(negative) deviations in $FM_t$. Similarly, $PEC_t$ and $NEC_t$ are generated from $EC_t$ variable. Replacing variables generated from Equations (2)–(5) by $FM_t$ and $EC_t$ in Equation (1), we may express our model as follows:

$$CO_t = f(Y_t, Y_t^2, PFM_t, NFM_t, PEC_t, NEC_t) \tag{6}$$

Now, we may have an asymmetric effect through different magnitudes or signs of coefficients of $PEC_t$, $NEC_t$, $PFM_t$ and $NFM_t$ on the $CO_t$ in the estimation of Equation (6). To regress the model in Equation (6), Figure 1 depicts the methodological framework. Testing the unit root problem may be considered a pre-condition to move the cointegration analysis. Therefore, each variable should be tested for a unit root problem.

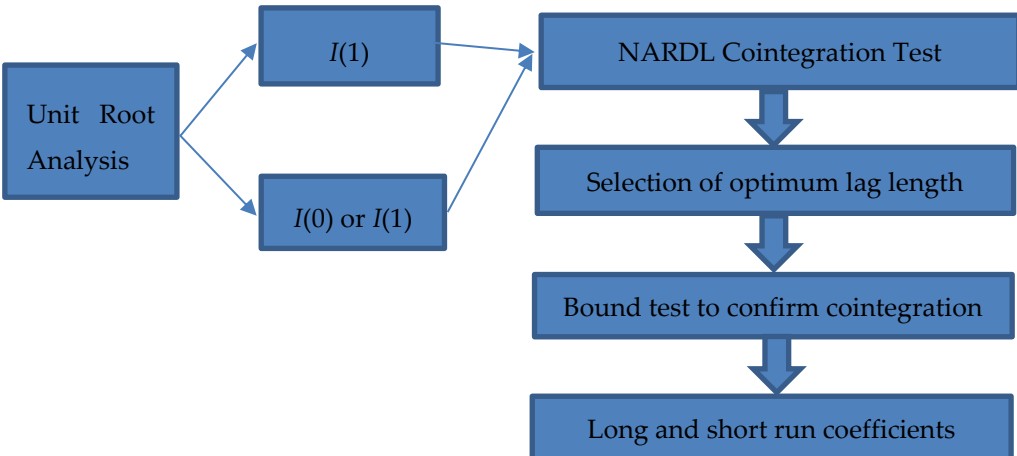

**Figure 1.** Methodological Framework.

Considering the first step of testing the unit root, we are using a test proposed by Ng and Perron [39] which examines the unit root problem on the detrended series ($x_t^d$) in the test Equation (7).

$$\Delta x_t^d = \eta_0 + \eta_1 t + \eta_2 x_{t-1}^d + \sum_{j=1}^{m} J_j \Delta x_{t-j}^d + \psi_t \tag{7}$$

$$MZ_a = [(x_T^d/T) - f_0]/2 \sum\nolimits_{t-2}^{T} (x_T^d)^2 * T^{-2} \tag{8}$$

$$MSB = \sqrt{\sum\nolimits_{t-2}^{T} (x_T^d)^2 * T^{-2}/f_0} \tag{9}$$

$$MZ_t = MZ_a * MSB \tag{10}$$

$$MZT = [\bar{c}^2 \sum\nolimits_{t-2}^{T} (x_T^d)^2 * T^{-2} + [(1-\bar{c})/T] * (x_T^d)^2]/f_0 \tag{11}$$

In the above equations, $f_0$ is a residual's range at 0 frequency. This test may be claimed as efficient due to the detrending procedure and due to its suitability in case of small-time sample size. The test utilizes the Equation (7) to test the unit root problem with $H_0$: $\eta_2 = 0$ and a rejection of $H_0$ may ensure the stationarity of $X_t$ variable. The null hypothesis is tested after choosing optimum lag length through Akaike Information Criterion (AIC) in Equation (7). Further, $H_0$ is tested with four modified statistics presented in Equations (8)–(11) to ensure the validity of the decision of stationarity. After that, we apply the Zivot and Andrews [40] test which incorporates a most significant structural break in the series while testing the unit root problem. This test is utilized with an objective to re-confirm the decision of stationarity of the series after considering a most significant breakpoint in series. Because a non-stationary series may be proved stationary if information of breakpoint is considered. The test equation is as follows:

$$\Delta x_t = \alpha_0 + \alpha_1 t + \alpha_2 DU_t^*(\lambda) + \alpha_3 DT_t^*(\lambda) + \alpha_4 x_{t-1} + \sum\nolimits_{j=1}^{m-1} \beta_j \Delta x_{t-j} + \xi_t \tag{12}$$

In Equation (12), $\lambda = T_B/T$ is to capture an optimum breakpoint ($T_B$) in the series to test the unit root problem. $DU_t^*(\lambda)$ may be assumed one and $DT_t^*(\lambda) = t - T\lambda$ in case of $t > T\lambda$ and zero otherwise. Then, this test verifies the unit root problem with a $H_0$: $\alpha_4 = 0$ considering one most significant break point in the series. A rejection of $H_0$ may ensure the stationarity of a series.

After doing the stationarity analysis mentioned above, testing the level of integration is very important. For example, all variables of our model are stationary at first difference $I(1)$ or these have mix order of integration with some variables level stationary $I(0)$ and others first difference stationary $I(1)$. If our dependent variable is $I(1)$ and independent variables have even a mixed order of integration, then we may move to cointegration analysis using Nonlinear Auto-Regressive Distributive Lag (NARDL) technique offered by Shin et al. [38]. Because the bound test of this technique is following lower bound with an assumption of $I(0)$ and upper bound with an assumption of $I(1)$ [41]. Therefore, cointegration analysis remains efficient even in the presence of a mixed order of integration. The test equation of NARDL of our model in Equation (6) is as follows:

$$\Delta CO_t = \gamma_0 + \gamma_1 CO_{t-1} + \gamma_2 Y_{t-1} + \gamma_3 Y_{t-1}^2 + \gamma_4 PFM_{t-1} + \gamma_5 NFM_{t-1} + \gamma_6 PEC_{t-1}$$
$$+ \gamma_7 NEC_{t-1} + \sum\nolimits_{j=1}^{q1} \delta_{1j} \Delta CO_{t-j} + \sum\nolimits_{j=0}^{q2} \delta_{2j} \Delta Y_{t-j} + \sum\nolimits_{j=0}^{q3} \delta_{3j} \Delta Y_{t-i}^2 + \sum\nolimits_{j=0}^{q4} \delta_{4j} \Delta PFM_{t-j} \tag{13}$$
$$+ \sum\nolimits_{j=0}^{q5} \delta_{5j} NFM_{t-j} + \sum\nolimits_{j=0}^{q6} \delta_{6j} \Delta PEC_{t-i} + \sum\nolimits_{j=0}^{q7} \delta_{7j} \Delta NEC_{t-i} + \xi_t$$

In Equation (13), the optimum lag length would be selected for q1, q2, q3, q4, q5, q6 and q7 using AIC by setting maximum two lags as we are utilizing the annual data of a period 1971–2014. Secondly, we may apply the bound test on a null hypothesis $\gamma_1 = \gamma_2 = \gamma_3 = \gamma_4 = \gamma_5 = \gamma_6 = \gamma_7 = 0$ to verify evidence of cointegration in the hypothesized model. To validate cointegration, the estimated F-value from the bound test may compare with efficient F-values of a small-time sample of Narayan [42]. Further, long run impacts may be captured with the help of normalized coefficients of lagged level variables normalized by the coefficient of $CO_{t-1}$. After that, we replace lagged level variables with Error Correction Term ($ECT_{t-1}$) in Equation (13), and short-run relation may be corroborated with a negative and significant coefficient of $ECT_{t-1}$. After that, short-run impacts may be explained by the estimated coefficients of differenced variables.

## 4. Results

The testing of the unit root problem may be considered the first step of any time series analysis. Therefore, we perform Ng and Perron [39] and Zivot and Andrews [40] assuming without and with a break respectively, and the results are presented in Table 2. The findings of Ng-Perron test show that at the level, unit root exists for all series and stationary at first difference. Zivot-Andrews test is applied to test whether a decision on non-stationary series may change after considering a break in analysis and this test suggests that $Y_t$ and $EC_t$ are showing stationary behavior at level which showed unit root before in Ng-Perron test. Also, stationarity is found at first differences for all variables in Zivot-Andrews test's results. Considering both unit tests' results, we may conclude a mixed order of integration which is sufficient for NARDL cointegration analysis. This cointegration technique may be applied if the dependent variable is $I(1)$ and others are $I(0)$ or $I(1)$.

**Table 2.** Unit root analyses.

| Ng and Perron [39] Test | | | | | |
|---|---|---|---|---|---|
| Variable | MZa | MZt | MSB | MPT | Decision |
| $CO_t$ | −13.7765(1) | −2.6137 | 0.1897 | 6.6769 | Non-Stationary |
| $Y_t$ | −9.2681(2) | −2.1068 | 0.2274 | 10.0180 | Non-Stationary |
| $FM_t$ | −6.3145(0) | −1.7477 | 0.2768 | 14.4226 | Non-Stationary |
| $EC_t$ | −2.7809(0) | −1.1282 | 0.4057 | 31.1944 | Non-Stationary |
| $\Delta CO_t$ | −20.7058(0) ** | −3.2054 | 0.1548 | 4.4746 | Stationary |
| $\Delta Y_t$ | −17.8135(0) ** | −2.9801 | 0.1673 | 5.1414 | Stationary |
| $\Delta FM_t$ | −20.2727(0) ** | −3.1838 | 0.1571 | 4.4950 | Stationary |
| $\Delta EC_t$ | −15.3313(0) * | −2.7683 | 0.1806 | 5.9463 | Stationary |
| Zivot and Andrews [40] Test | | | | | |
| | Level | | First Difference | | Decision |
| | t-stat | Break Year | t-stat | Break Year | |
| $CO_t$ | −4.1383(0) | 2003 | −7.6790(0) *** | 1995 | Stationary |
| $Y_t$ | −7.1584(0) *** | 1981 | −6.5853(8) *** | 1994 | Stationary |
| $FM_t$ | −3.0183(0) | 1974 | −6.7280(0) *** | 1983 | Stationary |
| $EC_t$ | −4.7076(0) ** | 1977 | -6.9523(0) *** | 1981 | Stationary |

Note: *, ** and *** are showing stationarity on 10%, 5% and 1% level of significance.

After performing integration analyses, we proceed to the cointegration analysis. The long and short runs' results of NARDL of Equation (13) are presented in Table 3. The Equation (13) is regressed after inclusion of dummy variable ($D_{2008}$) to look for the effect of the financial crisis of 2008. The known break year of financial crisis is preferred in analysis over other unknown break years identified by Zivot-Andrews test because of a reason that our major focus of this research is to inquire the environmental effect of FMD. Therefore, the financial crisis may be considered as important to shape the FMD-pollution relationship and to complete the information in the model. The regressed model may be claimed un-biased as F-values of heteroscedasticity, serial correlation and Ramsey RESET tests and Chi-square value of Jarque-Bera test are reasonably low, and their respective p-values are reasonably high enough to conclude that our estimated model is out of any econometric problem.

The Cumulative Sum (CUSUM) and CUSUM of squares tests displayed in Figure 2 are showing the stability and consistency of estimates. Moreover, our model is showing a cointegration as F-value from the bound test is found higher than upper critical F-value of Narayan [42] at 5% level of significance. Existence of a long and short run associations is also corroborated by estimated negative and statistically significant coefficient of $ECT_{t-1}$ and speed of convergence from a short disequilibrium may be considered less than two years as per estimated coefficient (−0.5823). After collecting all above evidence, we may shift our discussions towards long and short-run impacts.

**Table 3.** $CO_2$ emissions per capita model results.

| Variable | Parameters | S.E. | t-Statistic | *p*-Value |
|---|---|---|---|---|
| **Long Run** | | | | |
| $Y_t$ | 42.0697 | 17.1714 | 2.4500 | 0.0211 |
| $Y_t^2$ | −2.1244 | 0.8609 | −2.4677 | 0.0202 |
| $PFM_t$ | −0.0972 | 0.2185 | −0.4448 | 0.6600 |
| $NFM_t$ | −1.3126 | 0.5127 | −2.5602 | 0.0164 |
| Wald Test | | $\chi^2 = 9.4051$ | | 0.0022 |
| $PEC_t$ | 0.2468 | 0.1689 | 1.4611 | 0.1555 |
| $NEC_t$ | 3.2358 | 0.9138 | 3.5412 | 0.0015 |
| Wald Test | | $\chi^2 = 11.7536$ | | 0.0006 |
| $D_{2008}$ | 0.0797 | 0.1020 | 0.7810 | 0.4416 |
| Intercept | −205.2250 | 85.4685 | −2.4012 | 0.0235 |
| **Short Run** | | | | |
| $\Delta Y_t$ | 0.7824 | 7.2952 | 0.1072 | 0.9154 |
| $\Delta Y_{t-1}$ | −12.2384 | 6.6307 | −1.8457 | 0.0759 |
| $\Delta Y_t^2$ | 0.0068 | 0.3601 | 0.0189 | 0.9850 |
| $\Delta Y_{t-1}^2$ | 0.6363 | 0.3314 | 1.9202 | 0.0655 |
| $\Delta PFM_t$ | −0.0566 | 0.1251 | −0.4523 | 0.6546 |
| $\Delta NFM_t$ | −0.1864 | 0.2156 | −0.8645 | 0.3949 |
| $\Delta PEC_t$ | 0.1437 | 0.0999 | 1.4394 | 0.1615 |
| $\Delta NEC_t$ | 2.6315 | 0.5760 | 4.5685 | 0.0001 |
| Wald Test | | $\chi^2 = 17.7046$ | | 0.0000 |
| $D_{2008}$ | 0.0464 | 0.0591 | 0.7851 | 0.4392 |
| $ECT_{t-1}$ | −0.5823 | 0.1239 | −4.6993 | 0.0001 |
| **Diagnostics** | | | | |
| Bound Test | F-value = 4.5772 | Narayan's [42] Critical Bound F-values<br>At 1% (3.383–4.832)<br>At 5% (2.504–3.723) | | |
| $F_{Hetro}$ | | 1.3730 | | 0.2322 |
| $F_{Serial}$ | | 0.2580 | | 0.7746 |
| $F_{RESET}$ | | 1.9339 | | 0.1761 |
| $\chi^2_{Normal}$ | | 0.8979 | | 0.6383 |

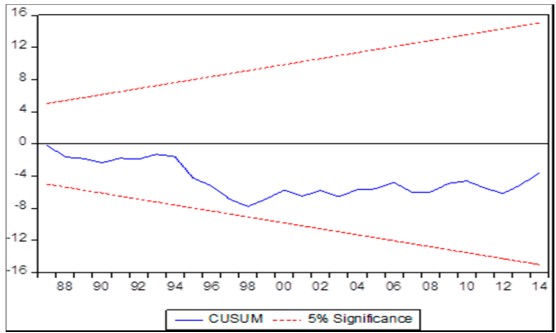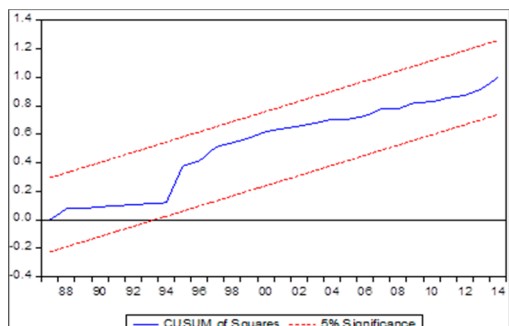

**Figure 2.** CUSUM and CUSUM of squares Tests.

In the long run results of Table 3, the coefficients of $Y_t$ and $Y_t^2$ are positive and negative respectively. It means that increasing GDP per capita has negative environmental effect before the turning point of the relationship. After turning point, it has a positive environmental effect. The overall result is corroborating an inverted U-shape relationship, and the EKC hypothesis holds. The turning point of quadratic effect of $Y_t$ on $CO_t$ may be estimated by taking exponent of a ratio

$(42.0697/-2(-2.1244))$, and the turning point is captured at GDP per capita = 19,961 constant US dollar approximately.

The coefficient of $PFM_t$ is negative but insignificant, and we may conclude that increasing FMD has no effects on $CO_2$ emissions per capita. However, the effect of $NFM_t$ is negative and significant. Further, Wald test on a null hypothesis ($H_0$) of the symmetric effect of $PFM_t$ and $NFM_t$ has been applied, and $H_0$ is rejected at 1% level. Therefore, it can be concluded the asymmetrical effects of FMD on the $CO_2$ emissions per capita. Further, the elastic coefficient of $NFM_t$ suggests that one percent decrease in the percentage of credit to GDP is responsible for 1.3126% increment of per capita $CO_2$ emissions. The impacts of $PEC_t$ is positive and statistically insignificant in the long run and lead to the conclusion that increasing energy consumption is not responsible for environmental degradation. Conversely, $NEC_t$ coefficient is positive and significant. Further, the effect of $NEC_t$ is found elastic and showing that one percent decreasing energy consumption per capita is helping in reducing 3.2358% of $CO_2$ emissions per capita. Moreover, Wald test on $H_0$ of the symmetric effect of $PEC_t$ and $NEC_t$ is rejected at 1% level. Therefore, it can be inferred that the above discussed the insignificant effect of $PEC_t$ and positive and significant effect of $NEC_t$ are corroborated by the Wald test as well. Also, the coefficient of $D_{2008}$ is statistically insignificant. Therefore, the financial crisis neither helps in reducing $CO_2$ emissions per capita nor is responsible for degradation in the environmental profile.

In the short run results, $\Delta Y_{t-1}$ and $\Delta Y_{t-1}{}^2$ show negative and positive respectively effects on the $CO_2$ emissions per capita. This result corroborates that short run lagged variable of income and income square are showing the U-shaped relationship. The short-run effects of positive and negative variables of FMD are showing insignificant results. So, Wald test is not applied to test asymmetry. It means that either increasing or decreasing FMD in the short run do not have any environmental effects. Further in the short run energy consumption effect, increasing per capita energy consumption has an insignificant effect and decreasing has a positive effect like long run results. The estimated elasticity of $NEC_t$ is slightly lower than long run elasticity but still is greater than one. One percent decrease in energy consumption per capita is helpful in reducing 2.6315% $CO_2$ emissions per capita in the short run. Further, short-run asymmetric effects of $PEC_t$ and $NEC_t$ are also corroborated by Wald test.

## 5. Discussions

The positive (negative) effects of $Y_t$ ($Y_t{}^2$) confirm the existence of the EKC hypothesis in our estimations in Saudi economy at the aggregate level. This statistically significant quadratic relationship corroborates the weakness in the model specification of the Alkhathlan and Javid [14] and Bekhet [15], which test the monotonic effect of income level on the $CO_2$ emissions. This finding is in contrast of Alshehry and Belloumi [17], who could not find EKC hypothesis in the transport sector of Saudi Arabia due to ignoring other sectors of the economy as transport sector is comprised less than 6% of GDP [43]. The right estimation of the EKC hypothesis is very important in the policy prospects of any country as economic growth with the sustainable environment is depending on the current position of a country. The turning point of the inverted U-shape curve is estimated at GDP per capita of 19,961 constant US dollars and the average GDP per capital of sample period is more than estimated turning point value. Therefore, we may conclude that Saudi Arabia is on the second phase of inverted U-shape relation. Moreover, further increasing GDP per capita may have a positive environmental effect as the second phase of EKC shows a negative relationship in economic growth and $CO_2$ emissions. Moreover, this result also corroborates that technique and composition effects are dominant on the scale effect of economic growth in this country. It can be evident from the increasing service sector contribution in national income as more than 50% of GDP has been generated from the service sector in the recent years and also through decreasing oil dependence as negative growth rates of oil sector were observed in the recent years [43]. In addition, government of Saudi Arabia should further focus on the service sector to ensure the sustainable environment and development by diversifying the economy further from oil dependence.

The effects of FMD on $CO_2$ emissions have been found asymmetrical in our findings. The effect of positive movements of FMD has been found insignificant which is in line of finding of Bekhet et al. [15] who report insignificant symmetrical effect of FMD on $CO_2$ emissions. This finding comfirms that increasing FMD, at least does not have adverse effects in terms of pollution emissions. The financial sector has shown faster growth in the recent year, i.e., more than 6% in 2017 [43] and this growth does not lead to any environmental consequences. Therefore, we suggest to expand the financial sector in the economy with an objective of diversification in the kingdom in her Vision 2030. Further, we find a negative relationship between the decreasing FMD and $CO_2$ emissions which is in line of symmetrical effect's finding of Salahuddin et al. [10] and Salahuddin et al. [37] for similar oil exporting countries GCC and Kuwait respectively and contrasts the insignificant effect of FMD reported by Bekhet et al. [15] for Saudi Arabia. Bekhet et al. [15] reported the symmetrical insignificant effects of FMD on $CO_2$ emissions which means that both increasing and decreasing FMD have insignificant effects but our results corroborate that the effect of increasing FMD is only insignificant. This finding highlights the importance of our testing asymmetrical effects of FMD on the $CO_2$ emissions. Moreover, our estimated negative elastic relationship between decreasing FMD and $CO_2$ emissions suggests that one percent decreasing FMD is responsible for environment degradation more than one percent. Therefore, government should support any negative growth in FMD to protect environment.

An asymmetrical relationship between energy consumption and $CO_2$ emissions has been estimated. The effect of increasing energy consumption has been found insignificant, corroborates the findings of Bekhet et al. [15]. But, positive and significant effects of decreasing energy consumption on $CO_2$ emissions rejects the insignificant symmetrical effect of energy consumption reported by Bekhet et al. [15] and is in line of the findings of Alkhathlan and Javid [14] and Alshehry and Belloumi [17]. Further, this positive relationship's elasticity is found at 3.24. Therefore, any negative movement in energy consumption would be helpful in reducing more than three times of $CO_2$ emissions. Therefore, the government should subsidize the energy saving technologies in both economic activities of production and consumption or subsidize the cleaner energy technology to reduce overall energy consumption in the economy for a better environment quality. Once we discuss the importance and application of cleaner technology, the role of social development should also be taken care by the government by realizing the aim of cleaner technology and cleaner ways of production and consumption and to increase energy efficiency with an ultimate objective of environment protection.

One of our objectives is to test the asymmetrical effects of FMD. Therefore, it seems pertinent to test the role of the global financial crisis of 2008 in affecting the pollution emissions. Our estimations corroborate that financial crisis has neither positive nor negative environmental effects. This result matches with sound economic conditions of Saudi Arabia during the global financial crisis. For example, real GDP grew at 6%, and exports grew at 34% in 2008 [43]. Therefore, the financial crisis did not affect the aggregate economy and hence pollution emissions.

## 6. Conclusions

Testing the EKC hypothesis and effect of FMD on $CO_2$ emissions are very important for the policy prospects of any country, but for Saudi Arabia, these could not be found significant in the past literature due to model specification issues. Therefore, the present study tries to improve the model specification by testing the asymmetrical effects of FMD and energy consumption per capita on the per capita $CO_2$ emissions along with testing the EKC hypothesis in Saudi Arabia. The testing asymmetrical effects is also scant in the empirical environmental literature. To achieve our objectives, unit root tests and NARDL have been applied on a maximum available data from 1971–2014. A mixed order of integration has been found in unit root analyses, and cointegration has been corroborated with the bound test. In the long run results, EKC hypothesis has been found significant in the relationship of income and pollutant emissions, and the turning point has been estimated at GDP per capita 19,961 constant US dollar. The asymmetrical effects of FMD have corroborated in the empirical findings in the long run

and increasing FMD has an insignificant effect on the $CO_2$ emission. On the other hand, decreasing FMD is found elastically responsible for environmental degradation. Therefore, we may conclude that decreasing FMD has adverse environmental effects. Further, the long run effects of energy consumption per capita are found asymmetrical and decreasing energy consumption per capita is found supportive of a clean environment by reducing pollution. In the short results, the lag of GDP per capita is seen to have a U-shape effect on $CO_2$ emissions per capita. The effect of FMD is found insignificant, and the effect of energy consumption is found asymmetrical. The effect of the financial crisis of 2008 is found insignificant both in long and short runs. In summing of, we contribute in the environment literature of Saudi Arabia by finding the significant EKC hypothesis at the aggregate level and the significant asymmetrical effects of FMD and energy consumption.

### 6.1. Policy Implication

The average GDP per capita of the whole sample period and in recent years found greater than the estimated turning point. Therefore, we may conclude that Saudi Arabia is in the second phase of EKC and increasing economic growth is not harmful to the environment because, Saudi Arabia has achieved the sufficient composition and technology effects. This may also be evident from the recent reduced oil dependence and increasing share of the services sector. Based on the result, we recommend the Saudi economy to focus on the services sector in her economic growth and diversification policies of Vision 2030 and to reduce oil dependence which may negative environmental effects.

Insignificant effect of increasing FMD suggests that the Saudi economy may enhance the share of financial sector without harming the environment. Further, FMD may also contribute to protecting the environment if government supports/subsidizes the financial sector in case of financing the cleaner technology projects. On the other hand, decreasing FMD is found have negative environmental effects. This leaves a point to ponder upon for the Saudi government to avoid negative growth shocks in financial market if it wants to sustain a certain level of environmental stability. For this purpose, central bank should support the financial markets in the negative growth periods and should provide anticipated information of financial market to avoid negative shocks and to ensure continues positive growth rates in this sector. Additionally, incorporation of energy efficient technologies in the energy segment of the country can be a way to protect the quality of the overall environment no matter which side FMD goes.

Lastly, we find that decreasing energy consumption has positive environmental effects. This outcome of reducing energy consumption cannot be achieved by actually cutting down on the consumption of energy in transport, production and other sector, but the ways energy is consumed can be improved that the same processes require a less amount of energy. However, to get to this point, the country will need more research in the energy segment to come up with energy efficient ideas and strategies both for the residential and commercial sectors. Further, the application of cleaner technology is very important in reducing the negative environmental effects of energy consumption.

### 6.2. Future Directions

To reduce negative environmental effects of energy consumption, economic growth or any other contributing agent, the most appropriate solution that comes to mind is the use of cleaner technology and to increase energy efficiency. However, there are many social and economic barriers in the way of application of cleaner technology because of lack of social immaturity in the society among many other reasons. Therefore, a mature level of social development is very necessary to implement clean environment policies. Further, cleaner technology may be expensive which may not be affordable by marginal social groups of society, and there may be many other dimensions in this context which may be counted as barriers in the way of implication for a clean environment. This present research was aimed at to single out the effect of FMD along with very necessary variables of economic growth and energy consumption using a limited available time sample. Therefore, the present research could not test the concept of social development or social barriers on the pollution emissions to avoid the

problems of the degree of freedom in the model. However, a future study may specifically target this issue by using different proxies of social development to test its impact on the environment in a single country case or a panel of countries.

**Supplementary Materials:** The following are available online at http://www.mdpi.com/1996-1073/11/12/3462/s1, Data.xlsx.

**Author Contributions:** Conceptualization, H.M. and A.S.A.; methodology, H.M.; software, H.M.; validation, H.M., M.F.; formal analysis, H.M.; investigation, A.S.A. and M.F.; data collection, A.S.A.; writing—original draft preparation, A.S.A. and M.F.; writing—review and editing, H.M. and M.F.; supervision, H.M.; project administration, H.M.

**Funding:** This research received no external funding.

**Conflicts of Interest:** The authors declare no conflict of interest.

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
