# Peer review of "Financial Market Development and Pollution Nexus in Saudi Arabia: Asymmetrical Analysis"

_energies, doi:10.3390/en11123462_

Round 1

Reviewer 1 Report

Dear Authors

In my opinion the theme of the article is very actual and interesting for the readers of the journal.

The manuscript under revision presents a very interesting model to assess information of ecological sustainability.

This research tests the EKC hypothesis in Saudi Arabia along with asymmetrical effects of FMD and energy consumption per capita on the per capita CO2 emissions.

From a long-term perspective, results show that EKC hypothesis has been found true in the relationship of income and pollutant emissions, and the turning point has been estimated at GDP per capita ~20,000 constant US dollar.

The authors also concluded that decreasing FMD has adverse environmental. Environmental stability prevalence when avoiding negative growth shocks in financial market.

The tested effect of the financial crisis has been found insignificant on CO2 emissions.

Additionally, the authors show that long run effects of energy consumption per capita are found asymmetrical and decreasing energy consumption per capita is found supportive of a clean environment.

From a long-term perspective, results show the lag of GDP per capita is seen to have a U-shape effect on CO2 emissions per capita and the effect of FMD is found insignificant.

The authors conclude that the decreasing energy consumption is helpful in reducing pollution, improving the ways of consume.

The paper is well structured, well written, the language is correct and clear, and the title and abstract clearly describe the content of the manuscript. However, in my opinion the authors must not put references in the Conclusion section.

In my opinion only minor revision is needed.

Best regards

Author Response

Review Report 1

In my opinion the theme of the article is very actual and interesting for the readers of the journal.

The manuscript under revision presents a very interesting model to assess information of ecological sustainability.

This research tests the EKC hypothesis in Saudi Arabia along with asymmetrical effects of FMD and energy consumption per capita on the per capita CO2 emissions.

From a long-term perspective, results show that EKC hypothesis has been found true in the relationship of income and pollutant emissions, and the turning point has been estimated at GDP per capita ~20,000 constant US dollar.

The authors also concluded that decreasing FMD has adverse environmental. Environmental stability prevalence when avoiding negative growth shocks in financial market.

The tested effect of the financial crisis has been found insignificant on CO2 emissions.

Additionally, the authors show that long run effects of energy consumption per capita are found asymmetrical and decreasing energy consumption per capita is found supportive of a clean environment.

From a long-term perspective, results show the lag of GDP per capita is seen to have a U-shape effect on CO2 emissions per capita and the effect of FMD is found insignificant.

The authors conclude that the decreasing energy consumption is helpful in reducing pollution, improving the ways of consume.

The paper is well structured, well written, the language is correct and clear, and the title and abstract clearly describe the content of the manuscript. However, in my opinion the authors must not put references in the Conclusion section.

 In my opinion only minor revision is needed.

Response:

Thank you very much for appreciation. Reference from conclusion section has been removed.

English language and style have been improved throughout the article. 

Reviewer 2 Report

Review of: Financial Market Development and Pollution Nexus in Saudi Arabia: Asymmetrical Analysis

Overview: this paper tests Environmental Kuznets Curve (EKC) hypothesis in Saudi Arabia along with asymmetrical effects of Financial Market Development (FMD) and energy consumption per capita on the per capita CO2 emissions. A mixed order of integration has been found in unit root analyses, and cointegration has been corroborated with the bound test. Finally, the authors have presented numerical results to show that the effect of FMD is found insignificant, and the effect of energy consumption is found asymmetrical and the decreasing energy consumption is found helpful in reducing pollution.

Recommendation: My suggestion is << Reconsider after major revision>>. Indeed, there are some points to consider:

1.      The contribution of the paper is not clear, thus, the authors shall to clarify the contribution in the introduction. What the novelty of your work compared to other published papers? How the paper contributions fill the gaps in the literature?

2.      The research status should briefly presented in the introduction. Authors present various works on carbon emissions but most of them are old work. To justify these issues, literature review should be improved by searching latest works regarding these issues. Thus, I suggest that authors should add more references from Energies Journal or other top journal and improve the literature review by latest references such as:  

Wang, Q., He, L., Zhao, D., & Lundy, M. (2018). Diverse Schemes of Cost Pooling for Carbon-Reduction Outsourcing in Low-Carbon Supply Chains. Energies, 11(11), 3013.

Turki, S., Sauvey, C., & Rezg, N. (2018). Modelling and optimization of a manufacturing/remanufacturing system with storage facility under carbon cap and trade policy. Journal of Cleaner Production.

3.      Please write the symbols in a table (Notation) and mentioned the decisions variables.

4.      Please explain how the results are obtained.

5.      The authors presented numerical results that are very interesting. Therefore, to improve the relevancy of the paper, the authors shall explaining the usefulness of these results in practice.

6.      Please make sure your conclusions' section undersc ore the scientific value added of your paper, and/or the applicability of your findings/results. Basically, you should enhance your findings, limitations, underscore the scientific value added of your paper, and/or the applicability of your contributions/shortages and future study in this session.

Author Response

Review Report 2

Review of: Financial Market Development and Pollution Nexus in Saudi Arabia: Asymmetrical Analysis

Overview: this paper tests Environmental Kuznets Curve (EKC) hypothesis in Saudi Arabia along with asymmetrical effects of Financial Market Development (FMD) and energy consumption per capita on the per capita CO2 emissions. A mixed order of integration has been found in unit root analyses, and cointegration has been corroborated with the bound test. Finally, the authors have presented numerical results to show that the effect of FMD is found insignificant, and the effect of energy consumption is found asymmetrical and the decreasing energy consumption is found helpful in reducing pollution.

Recommendation: My suggestion is << Reconsider after major revision>>. Indeed, there are some points to consider:

1. The contribution of the paper is not clear, thus, the authors shall to clarify the contribution in the introduction. What the novelty of your work compared to other published papers? How the paper contributions fill the gaps in the literature?

Response:

First of all, I would like to thank you to provide us very useful comments which significantly improved the quality of our paper.

To answer the question 1, Two paragraphs have been added in the Introduction section (lines 59-87). In which, one paragraph in lines (59-69) discussed all related literature in MENA and GCC regions because of Saudi Arabia’ location. To clarify the present research contribution in Saudi literature, all related literature in Saudi Arabia is discussed in a way to highlight research gap and research contribution is also presented in last paragraph (lines 70-87).

Please Note: Introduction section has been divided into two separate sections (1. Introduction and 2. Literature Review) due to comment # 1 of 3rd reviewer. Further, literature gap has also been discussed in section 2, last paragraph (lines 213-226) with support of two above paragraphs on Saudi Literature in (lines 191-202).

2. The research status should briefly presented in the introduction. Authors present various works on carbon emissions but most of them are old work. To justify these issues, literature review should be improved by searching latest works regarding these issues. Thus, I suggest that authors should add more references from Energies Journal or other top journal and improve the literature review by latest references such as:

Wang, Q., He, L., Zhao, D., & Lundy, M. (2018). Diverse Schemes of Cost Pooling for Carbon-Reduction Outsourcing in Low-Carbon Supply Chains. Energies, 11(11), 3013.

Turki, S., Sauvey, C., & Rezg, N. (2018). Modelling and optimization of a manufacturing/remanufacturing system with storage facility under carbon cap and trade policy. Journal of Cleaner Production.

Response:

The literature Review has been updated by incorporation of 7 recent studies and presented in section 2 including the suggested studies above, suggested studies by comment # 8 of 3rd reviewer and some other studies from energies journal in the lines (102-136). These references have been updated in (lines 570-586).

3. Please write the symbols in a table (Notation) and mentioned the decisions variables.

Response:

In the response of this comment and also following comment # 4 of 3rd reviewer, a separate section 3.1 has been added and table 1 has been added to explain all variables’ notation and descriptions in (lines 229-234). Further, decision of stationarity has been added in the table 2 adding 6th column in (lines 328-329).

4. Please explain how the results are obtained.

Response:

Section 3.2 has been added to explain in detail the methodology to produce the results in (lines 236-316). Particularly, new text and equations are added, which was absent before, to extend the clarify of the methodology of results in (lines 255-305) and supporting reference is added in (lines 605-606). 

5. The authors presented numerical results that are very interesting. Therefore, to improve the relevancy of the paper, the authors shall explaining the usefulness of these results in practice.

Response:

In the response of this comment and also following comment # 7 of 3rd reviewer, a separate section # 5 has been added to improve the discussion of results by caring the usefulness of results in (lines 387-445). 

6. Please make sure your conclusions' section underscore the scientific value added of your paper, and/or the applicability of your findings/results. Basically, you should enhance your findings, limitations, underscore the scientific value added of your paper, and/or the applicability of your contributions/shortages and future study in this session.

Response:

In response this comment, Text has been added to clarify the objective and significant of study in conclusion section (lines 447-453), contribution mentioned in lines (466-468). A separate section 6.1 has been extended to discuss policy relevance in (lines 470-495) and section 6.2 has been added for future directions in (lines 496-510 which also cares the comment # 8 of reviewer 3).

English language and style have been improved throughout the article. 

Reviewer 3 Report

 The paper presents very interesting research which fits to the scope of the journal, however, there are some basic disadvantages which should not appear in scientific article.

1. The introduction section is too long and should be more concise. The introduction should briefly present the context and the scientific background. Detailed review of other studies should be moved to a separate section, e.g., a literature review section.

2. Based on the literature review, there should be clearly defined a gap in the current state-of-art.

3. Gap in knowledge should lead to define clear aim of the research. Explicit aim of the research allows readers to verify in the end of the paper if the goal has been achieved. The aim of the research should appear in the main text – not only in abstract.

4. The paper fails to characterize input data. Separate section should focus on database that was used for the study, the accuracy of the data (is the collection method the same for the whole period?), the spatial distribution of the data (is whole country aggregated into one information per each year or is it divided into smaller units?).

5. The “Methods” section should avoid introductions that are not necessary for methods. Current version makes the methods blurred. In order to improve the visibility of the method I suggest to present the methodological framework in diagram, and then describe each step.

6. The description of variables in equations should be listed below the formula – almost like in a bullet points, not in the body text.

7. Results and Discussion should be separated. Obtained results should be also better discussed in order to practice. What are the implications? How the current situation influence socio-environmental system? Which changes (if any) are needed to improve the stability of the system?

8. I have a feeling that the Authors strongly focus on the relation between energy consumption and CO2 emissions (Introduction in lines: 54, 64, 67-68, 92; second paragraph in Conclusions), however, very little focus is devoted to shifting into cleaner energy. Technological aspects of clean energy are mentioned only two times (Introduction in lines: 95, 104; omitted in Conclusions), and there is no remark concerning social barriers in application of renewable energy in the society, especially in excluded social groups. As the intention of the Authors was to refer to the Environmental Kuznets Curve, the aspect of development should refer to the higher development status of the country. On the one hand, that development is strongly related to the economic situation and the ability of new technologies application (more environmental friendly solutions). On the other hand, once the new technology is accessible at the local market, there is a social obstacle connected with convincing citizens to apply it. Therefore, the aspect of social development is also very important variable in EKC approach, however, this is totally omitted in the current version of the paper. It might be valuable to enrich the paper with examples from highly developed countries, especially from renewable energy applications in socially vulnerable groups, like poorer or older people, see for instance: “A multi-case study of innovations in energy performance of social housing for older adults in the Netherlands” Energy and Buildings 158 (2018) 1762–1769; “Social housing and renewable energy: Community energy in a supporting role” Energy Research & Social Science 38 (2018) 110-113.

Author Response

Review Report 3

The paper presents very interesting research which fits to the scope of the journal, however, there are some basic disadvantages which should not appear in scientific article.

1. The introduction section is too long and should be more concise. The introduction should briefly present the context and the scientific background. Detailed review of other studies should be moved to a separate section, e.g., a literature review section.

Response:

First of all, I would like to thank you to provide us very useful comments which significantly improved the quality of our paper.

To answer the question 1, Introduction section has been divided into two separate sections (1. Introduction and 2. Literature Review). Introduction section has briefly discussed the scientific background of the topic in context of MENA, GCC and KSA in last two paragraphs in (lines 59-87) to highlight the contribution of the study.

2. Based on the literature review, there should be clearly defined a gap in the current state-of-art.

Response:

Literature gap has been discussed in section 2, last paragraph (lines 213-226) with support of two above paragraphs on Saudi Literature in (lines 191-202).

3. Gap in knowledge should lead to define clear aim of the research. Explicit aim of the research allows readers to verify in the end of the paper if the goal has been achieved. The aim of the research should appear in the main text – not only in abstract.

Response:

Aims of the study have been discussed in Introduction section in (lines 84-87) and in Literature Review section in (lines 220-226) and in the conclusion section in (lines 447-453). The achievement of goals has been discussed in conclusion section in (lines 466-468).

4. The paper fails to characterize (describe) input data. Separate section should focus on database that was used for the study, the accuracy of the data (is the collection method the same for the whole period?), the spatial distribution of the data (is whole country aggregated into one information per each year or is it divided into smaller units?).

Response:

Following this comment and your comment # 6, A separate section # 3.1 has been added along with a table 1 with all relevant details in (lines 229-234).

5. The “Methods” section should avoid introductions that are not necessary for methods. Current version makes the methods blurred. In order to improve the visibility of the method I suggest to present the methodological framework in diagram, and then describe each step.

Response:

To answer this comments, introduction of model has been shifted in Introduction section in (lines 32-38 and 42-58). To improve the methodology, Figure 1 has been incorporated as suggested and details of methodology step by step have been improved in (lines 256-316) and particularly, text is added to clear all missing discussions of methodology in last version of paper in (256-305) along with necessary equations and references.

6. The description of variables in equations should be listed below the formula – almost like in a bullet points, not in the body text.

Response:

In the response of this comment and also following comment # 3 of 2nd reviewer, a table # 1 has been developed and added to list the variables with maximum description in (line 234). 

7. Results and Discussion should be separated. Obtained results should be also better discussed in order to practice. What are the implications? How the current situation influence socio-environmental system? Which changes (if any) are needed to improve the stability of the system?

Response:

Results and Discussion sections have been separated into sections # 4 and 5 respectively. The discussions of results with implications and suggested points have been added in detail with support of facts extracted from newly added reference # 43 in (lines 387-445).

8. I have a feeling that the Authors strongly focus on the relation between energy consumption and CO2 emissions (Introduction in lines: 54, 64, 67-68, 92; second paragraph in Conclusions), however, very little focus is devoted to shifting into cleaner energy. Technological aspects of clean energy are mentioned only two times (Introduction in lines: 95, 104; omitted in Conclusions), and there is no remark concerning social barriers in application of renewable energy in the society, especially in excluded social groups. As the intention of the Authors was to refer to the Environmental Kuznets Curve, the aspect of development should refer to the higher development status of the country. On the one hand, that development is strongly related to the economic situation and the ability of new technologies application (more environmental friendly solutions). On the other hand, once the new technology is accessible at the local market, there is a social obstacle connected with convincing citizens to apply it. Therefore, the aspect of social development is also very important variable in EKC approach, however, this is totally omitted in the current version of the paper. It might be valuable to enrich the paper with examples from highly developed countries, especially from renewable energy applications in socially vulnerable groups, like poorer or older people, see for instance: “A multi-case study of innovations in energy performance of social housing for older adults in the Netherlands” Energy and Buildings 158 (2018) 1762–1769; “Social housing and renewable energy: Community energy in a supporting role” Energy Research & Social Science 38 (2018) 110-113.

Response:

These are really excellent comments. In my opinion, a separate study should be conducted on the social development indicators and clean energy consumption and/or pollution emissions relationships. The present study could not be able to incorporate this/these variables in the model due to limited available time-sample data to avoid a problem of degree of freedom in the estimated model. However, a separate study may target only this objective by using a long time period of concerned country or by working on a panel of group of similar countries. Your suggestion has been reflected in the newly added section # 6.2 future direction in (lines 496-510) to conduct a future study on this issue.

Additionally, suggested literature has been discussed in the literature review section with prospect of social issues in (lines 115-124) and further social development and barrier discussions in the way of cleaner technology have been added in various sections of study in (lines 28 and 436-439 also discussed in mentioned before portions in lines 115-124 and 496-510).     

Round 2

Reviewer 1 Report

In my opinion, the submitted manuscript is ready to be published.

Reviewer 2 Report

The manuscript is improved and the authors have provided significant changes, the related literature and the conclusion are well presented.   I recommend the acceptance of the paper in the current form

Reviewer 3 Report

The paper has been corrected according to my previous suggestions and can be published in its current form.